Quantitative analysis of vegetation restoration and potential driving factors in a typical subalpine region of the Eastern Tibet Plateau

Feng Yu 1
Wang Juan id.wangjuan@foxmail.com 2
Zhou Qin 3
Bai Maoyang 1
Peng Peihao 1 2
Zhao Dan 4
Guan Zengyan 2
Liu Xian’an 5
1 College of Earth Sciences, Chengdu University of Technology , Chengdu , China
2 College of Tourism and Urban-Rural Planning, Chengdu University of Technology , Chengdu , China
3 Chengdu OCI Medical Devices Co., Ltd , Chengdu , China
4 School of Tourism and Culture Industry, Sichuan Tourism University , Chengdu , China
5 College of Art, Sichuan Tourism University , Chengdu , China
Li Chenxi
Electronic publication date: 2022 Apr 28
Publication date: 2022
Volume: 10
Electronic Location ID: e13358
Received 2021 Nov 26; Accepted 2022 Apr 8
Copyright: ©2022 Feng et al.
Copyright year: 2022
Copyright holder: Feng et al.
License: This is an open access article distributed under the terms of the Creative Commons Attribution License, which permits unrestricted use, distribution, reproduction and adaptation in any medium and for any purpose provided that it is properly attributed. For attribution, the original author(s), title, publication source (PeerJ) and either DOI or URL of the article must be cited.
License URL: https://creativecommons.org/licenses/by/4.0/

Keywords: FVC, Vegetation restoration, Driving factors, Eastern Tibet Plateau, Subalpine

Funding: The Biodiversity Survey and Assessment of the Shaluli Mountain System 80303-AZS023 The National Key R&D Program of China 2017YFC0505001 This research was funded by the Biodiversity Survey and Assessment of the Shaluli Mountain System (No. 80303-AZS023) and the National Key R&D Program of China (No. 2017YFC0505001). The funders had no role in study design, data collection and analysis, decision to publish, or preparation of the manuscript.

==============================
Vegetation restoration is an essential approach to re-establish the ecological balance in subalpine areas. Changes in vegetation cover represent, to some extent, vegetation growth trends and are the consequence of a complex of different natural factors and human activities. Microtopography influences vegetation growth by affecting the amount of heat and moisture reaching the ground, a role that is more pronounced in subalpine areas. However, little research is concerned with the characteristics and dynamics of vegetation restoration in different microtopography types. The respective importance of the factors driving vegetation changes in subalpine areas is also not clear yet. We used linear regression and the Hurst exponent to analyze the trends in vegetation restoration and sustainability in different microtopography types since 2000, based on Fractional Vegetation Cover (FVC) and identified potential driving factors of vegetation change and their importance by using Geographical Detector. The results show that: (1) The FVC in the region under study has shown an up-trend since 2000, and the rate of increase is 0.26/year (P = 0.028). It would be going from improvement to degradation, continuous decrease or continuous significant decrease in 47.48% of the region, in the future. (2) The mean FVC is in the following order: lower slope (cool), lower slope, lower slope (warm), valley, upper slope (warm), upper slope, valley (narrow), upper slope (cool), cliff, mountain/divide, peak/ridge (warm), peak/ridge, peak/ridge (cool). The lower slope is the microtopographic type with the best vegetation cover, and ridge peak is the most difficult to be afforested. (3) The main factors affecting vegetation restoration in subalpine areas are aspect, microtopographic type, and soil taxonomy great groups. The interaction between multiple factors has a much stronger effect on vegetation cover than single factors, with the effect of temperatures and aspects having the most significant impact on the vegetation cover changes. Natural factors have a greater impact on vegetation restoration than human factors in the study area. The results of this research can contribute a better understanding of the influence of different drivers on the change of vegetation cover, and provide appropriate references and recommendations for vegetation restoration and sustainable development in typical logging areas in subalpine areas.

Introduction

Vegetation is considered a critical factor in global terrestrial ecosystem changes (Ge et al., 2021; Kelly, Tuxen & Stralberg, 2011). It is an essential contributor to regulating the carbon cycles balance, reducing greenhouse gases, and mitigating climate change (Emamian et al., 2021; Hu et al., 2010), and reflects the fundamental characteristics of geological, geomorphological, climatic, hydrological, and soil (Newbold et al., 2015; Xu, Wang & Yang, 2017). Nowadays, environmental degradation has become a globally recognized topic of attention. China was identified to be amongst the countries with the worst ecological degradation, suffered vegetation degradation, soil erosion, and desertification (Yu et al., 2021). The ecological restoration was a crucial factor for re-establishing ecological balance and reversing environmental deterioration (Ma, Lv & Li, 2013), in which vegetation played a vital part (Zhou et al., 2021). It is critical to understand the vegetation cover dynamics and the drivers for policymaking by the administration.

The Tibetan Plateau (TP) is extremely vulnerable in terms of forest management and deforestation due to extreme climatic conditions and ecosystem fragility (Zhang et al., 2012). Over the past decades, extensive zones of deforestation have been left on the Eastern Tibetan Plateau (ETP), due to logging-related destruction (Xiong et al., 2021). Since 2000, China has implemented a series of ecological projects, such as the Natural Forest Protection Project (NFPP) (Mullan et al., 2010), and the Return of Cropland to Forests and Grasses Project (RCFGP) (Xie et al., 2006). However, ecosystems in this region are sensitive to climate change (Xiaodan, Genwei & Xianghao, 2011), and natural recovery of vegetation is a difficult and slow process. Therefore, there is a requirement to monitor vegetation dynamics and the efficacy of vegetation restoration to facilitate timely intervention and management by researchers and policymakers.

The Normalised Difference Vegetation Index (NDVI) is sensitive to the spectral information of vegetation and has been widely used in vegetation monitoring (Tang, He & Li, 2020). But it also has limitations, with sparsely vegetated areas having high canopy background signal noise and high vegetation areas having low saturation (Anees et al., 2022). The FVC calculated by NDVI reduces the effect of these limitations, effectively reducing the uncertainty caused by the spectral characteristics of unvegetated areas (He, Shi & Fu, 2021). It represents vegetation growth trends to some extent (Gao et al., 2017b; Wen et al., 2013), is widely used in studies related to climate change and vegetation restoration (He, Shi & Fu, 2021; Yan et al., 2011; Zhou, Shangguan & Zhao, 2006).

Vegetation change is influenced by a combination of natural factors and human activities (Liu et al., 2021; Liu et al., 2018), and it has complex drivers (Chen et al., 2020b; Zhu, Baskonus & Gao, 2020). The combination of various drivers determines vegetation patterns and regional trends (Boschetti et al., 2013; Chen et al., 2016). Considerable research has revealed that climate, topography, and soils were the major factors influencing vegetation change (Chen et al., 2020b; Peng, Kuang & Tao, 2019). Liu et al. (2021) in the northeastern TP showed that mean annual temperature, soil type, and elevation were the dominant factors driving vegetation change. In addition, human activities were another important factor affecting vegetation growth (Meng et al., 2019). Its impact on the ETP has gradually increased in the last decade (Chen, Yan & Lu, 2020a). However, fewer studies have analyzed the influence of topography on vegetation restoration in subalpine areas, especially the association between vegetation cover and microtopography types. The respective importance of these various drivers for vegetation change in the ETP subalpine region is not yet clear.

The influence of different drivers on the role of vegetation change and the spatial heterogeneity of factors has been neglected in prior research (Liu et al., 2021). Geographic Detector (Geodetector) can fill these gaps, and this spatial statistical approach allows for an integrated analysis of the different factors influencing FVC variation (Wang et al., 2010). It considers the spatial heterogeneity of factors and can quickly determine the importance of each factor, thereby determining the explanatory power of individual factors and the synergistic effects of multiple factors (Song et al., 2020). In this study, Geodetector was used to identify the main drivers of vegetation restoration, which can improve the efficiency of the ecological projects and provide valuable references for ecological management.

Therefore, to explore the factors affecting vegetation change in subalpine areas, we chose the Muru Basin, a typical subalpine deforestation and restoration area located on ETP, as our study area. Firstly, we used Google Earth Engine cloud platform to calculate and derive the FVC for 2000–2020; then we calculated the slope of FVC change and the Hurst exponent to analyze the spatial and temporal variation characteristics of FVC and change patterns on different microtopographic types; finally, we used the Geodetector to measure the contribution of natural and human factors to FVC change. In general, the aims of this research were (1) to analyze the spatial and temporal changes in FVC for 2000-2020 and their future development; (2) to identify differences in FVC responses between microtopographic types; (3) to identify the main factors affecting vegetation variation and evaluate the specific effects of each factor on vegetation restoration. The innovations of this study are mainly (1) to make up for the relatively few previous studies on microtopography types on vegetation cover changes; (2) to identify the main drivers of vegetation restoration and provide theoretical references for vegetation restoration in subalpine logging areas.

Materials & Methods

Study area

Our study area is sited in the Muru Basin (30.6342°∼30.8132°N, 101.1160°∼101.2772°E), Daofu County, Garze Tibetan Autonomous Prefecture, Sichuan Province, on the ETP (Fig. 1). It has an area of approximately 268.40 km2, the elevation range is 2,860∼4,932 m. This area experienced a long period of logging until the implementation of the NFPP and RCFGP around 2000, and the vegetation in this area was gradually restored. The species types in this area were essentially the same before and after the restoration, as native species were used for all vegetation restoration. The field survey found that the study area is dominated by subalpine evergreen coniferous forests, with the main species being Picea likiangensis and Abies squamata, mixed with Betula albosinensis and Populus davidiana at lower altitudes regions, with a clear community stratification, the main species in the shrub layer are Rhododendron decorum, Lonicera tangutica, Sorbus rehderiana, and Rosa sweginzowii. Some areas of evergreen coniferous forest have degraded into scrub after logging, with the main established species being Rhododendron decorum, Rhododendron lapponicum, Quercus monimotricha, etc. According to the nearest Daofu meteorological station data (101.11667°E, 30.98333°N; 2957.2 m a.s.l, 1982-2018), the average annual temperature is 8.23 °C. The warmest month is July, with an average temperature of 16.12 °C. The coldest month is January, with an average temperature of −1.51 °C. The land cover types are mainly forests, shrublands, and meadows. The majority of the basin is forest and shrub, with meadows mainly found at higher elevations along the basin margins. Traces of human activities such as cultivated land, roads, and residential areas are mainly located in the valleys. Soil taxonomy great groups (USDA system, https://www.openlandmap.org) mainly include Cryoboralfs, Cryoborolls, Argicryolls, Haplustalfs, and Cryumbrepts.

Figure 1 Location of Muru Basin in Eastern Tibet Plateau.

The map is reproduced from Tianditu (http://www.tianditu.gov.cn).

Data sources

Landsat satellite images at 30 m spatial resolution were obtained from Google Earth Engine (GEE, https://developers.google.com/earth-engine/), which provides uninterrupted global multispectral surface imagery every 16 days. We used Landsat 5 from 2000 to 2011, with 178 images, and Landsat 7 in 2012, with 14 images, and Landsat 8 from 2013 to 2020, with 145 images. All images were radiometrically calibrated, atmospherically corrected, and geo-referenced to reduce interference and improve image quality. In addition, cloud masks were used for all images to eliminate the effect of clouds. We used median calculations to composite each year’s images into one image without clouds in the GEE. This method is more resistant to extreme values and can be representative of the period studied (Bunting, Munson & Bradford, 2019).

Elevation, slope, and aspect were generated from DEM at 30 m spatial resolution obtained via the Geospatial Data Cloud (http://www.gscloud.cn/). The microtopographic type uses the 90 m resolution Global ALOS landforms dataset (https://www.sciencebase.gov/catalog/item/564b4bb0e4b0ebfbef0d31d2), which takes full consideration of Continuous Heat-Insolation Load Index and the multi-scale Topographic Position Index (Theobald et al., 2015), the study area has been classified into 13 microtopographic types (lower slope, lower slope (cool), lower slope (warm), upper slope, upper slope (cool), upper slope (warm), valley, valley (narrow), cliff, mountain/divide, peak/ridge (warm), peak/ridge (cool), and peak/ridge).

Soil taxonomy great groups, soil pH, and soil water content with a spatial resolution of 250 m were obtained from OpenLandMap (https://www.openlandmap.org/). Soil taxonomy great groups use the Soil Texture Class (USDA system) at 0 cm depth, soil pH uses the soil pH in H2O at 0 cm depth, and soil water content at 0 to 200 cm uses all bands in the Soil water content dataset at 33 kPa (field capacity).

Climatic datasets were obtained from WorldClim version 2.1 (https://www.worldclim.org/data/worldclim21.html), released in January 2020, spatial resolution is 30 s. The datasets included monthly values of the minimum temperature, maximum temperature, average temperature, precipitation, solar radiation, wind speed, and water vapor pressure. Based on these dataset, the annual average temperature, annual maximum temperature, annual minimum temperature, annual precipitation, and annual average water vapor pressure were calculated.

In addition to natural factors such as climate, topography, and soil, vegetation recovery is also influenced by human activities (Zhou et al., 2021). Roads and residences are the concrete embodiment of human activities in the study area, so we used distance to roads and residences to quantify the intensity of human activity, other studies have also used these two indicators (Liu et al., 2021). Roads and residences information was obtained manually in Google Earth Pro, then the distances of roads and distance to residences were calculated in ArcGIS 10.7 using the euclidean distance tool and finally resampled to a spatial resolution of 30 m.

To avoid serious multicollinearity, the Variance Inflation Factor (VIF) was calculated in ArcGIS 10.7 using the Ordinary Least Squares tool. Usually, VIF > 10 implies the possibility of serious multicollinearity (Marcoulides & Raykov, 2019). The potential driving factors with VIF > 10 were excluded; the final 12 factors were applied to the next step of the analysis (Table 1).

Table 1 Potential Driving Factor and VIF.

Category	Variable	Source	Spatial resolution	VIF	
Topography	Aspect	Geospatial data cloud
(https://www.gscloud.cn/)	30 m	1.0869	
Slope		1.1513	
Microtopographic type	Theobald et al. 2015
(https://www.sciencebase.gov/catalog/item/564b4bb0e4b0ebfbef0d31d2 )	90 m	1.1213	
Soil	Soil taxonomy great groups	OpenLandMap
(https://www.openlandmap.org)	250 m	1.3425	
Soil pH	2.7000	
Soil water content at 0 cm	8.0044	
Soil water content at 10 cm	9.4203	
Soil water content at 30 cm	9.1285	
Soil water content at 200 cm	5.5598	
Climate	Annual average temperature	WorldClim version 2.1
(https://www.worldclim.org/data/worldclim21.html	30 s	7.0501	
Human activities	Distance to roads	Google Earth Pro	30 m	2.2141	
Distance to residences	30 m	4.6635	

Vegetation cover calculation

Plant leaves have different absorption and reflectivity of red and near-infrared light wavelengths (Xu et al., 2019). The multi-spectral sensor’s spectrum bands contain both visible and infra-red wavelengths, which combine to produce vegetation indices (Brown et al., 2006; Wang, Liu & Huete, 2003). NDVI is calculated as the proportion of the difference between the near-infrared and red bands to the sum (Bianchi, Villalba & Solarte, 2019). NDVI ranges between −1.0 to +1.0, where a value less than 0 corresponds to without vegetation and greater than 0 corresponds to vegetation (Brehaut & Danby, 2018). The FVC is the proportion of area with vegetation cover to the whole area (Hu et al., 2020). The FVC is usually calculated from NDVI data in an image element dichotomous model with the following equation (He, Shi & Fu, 2021): (1) FVC=NDVI−NDVInNDVIv−NDVIn

where NDVI is the NDVI value of the image pixel, NDVIv and NDVIn represent the NDVI values of the vegetation and bare ground image pixel, respectively. In the actual calculation, the NDVI values of 5% and 95% of the cumulative frequency of the histogram are taken as NDVIn and NDVIv. To avoid the FVC results being outside the range of 0 to 1, the FVC was set to 0 for pixels with NDVI less than or equal to NDVIn and 1 for pixels with NDVI greater than or equal to NDVIv.

FVC change trend and sustainability analysis

Linear regression equation fitting methods can accurately measure the spatial patterns of dynamic changes in FVC and identify trends in individual pixels. This study used this method to simulate spatial trends in FVC from 2000∼2020. The formula is as follows He, Shi & Fu (2021) and Zhang, Ge & Zhang (2020): (2) SLOPE=n×∑i=1ni×FVCi− ∑i=1ni∑i=1nFVCi/n×∑i=1ni2−∑i=1ni2

where SLOPE is the trend of FVC, n is the number of years in a time series, i is the annual variable. SLOPE > 0 indicates a tendency for vegetation cover to improve, while the opposite implies a tendency to deteriorate. For determining the significance of FVC tendency, a T-test has been applied and per pixel’s P-value was computed. The results were classified into five categories: extremely significant increase (SLOPE > 0, P < 0.01), significant increase (SLOPE > 0, 0.01 <P < 0.05), extremely significant decrease (SLOPE < 0, P < 0.01), significant decrease (SLOPE < 0, 0.01 < P < 0.05), and insignificant change (P > 0.05).

The Hurst exponent is a useful approach to predict future trends in FVC (Emamian et al., 2021). The Hurst exponent utilizes the benefits of self-covariance and is usually applied at measuring the stationarity of big data sequences in the natural world through the Rescaled Range Series Analysis (Hurst, 1951; Zhang, Ge & Zhang, 2020). Based on the FVC from 2000 to 2020, the Hurst exponent has been adopted in our research to predict the future tendency. Provided a series of time sequences xi, i = 1, 2, 3, …, n, the Hurst exponent was computed as follows Emamian et al. (2021):

First, split the time series into different segments.

Second, calculate the mean value of each segment: (3) m=1n∑i=1nxi

Third, calculate the series of deviations: (4) yi=xi−m

Fourth, calculate the widest difference: (5) Ri=maxy1,y2,y3,…,yi−miny1,y2,y3,…,yi

Finally, calculate the standard deviation: (6) Si=1n∑i=1nxi−m212

With an increase in i, various values of R/S were derived and the below expressions were obeyed: (7) Ri/Si∝iH.

In which H is the Hurst exponent. 0 < H < 0.5 means a tendency for future variation to be the reverse direction of the history (opposite); H = 0.5 indicates random fluctuation (random); 0.5 < H < 1 represents a tendency for future variation is same as the history (continuous).

Combining the values of the SLOPE and H, we have divided the future trends of FVC into six groups: random fluctuation (H = 0.5), continuous decrease (0.5 < H < 1, SLOPE < 0, P > 0.05), continuous increase (0.5 < H < 1, SLOPE > 0, P > 0.05), continuous significant decrease (0.5 <H < 1, SLOPE < 0, P < 0.05), continuous significant increase (0.5 < H < 1, SLOPE > 0, P < 0.05), from degradation to improvement (0 < H < 0.5, SLOPE < 0), and from improvement to degradation (0 < H < 0.5, SLOPE > 0).

Geographical detector

Geodetector (http://www.geodetector.cn/) incorporates a range of spatial statistical methodologies, which explore the explanatory variables affecting the dependent variable through a spatial variance analysis (Wang, Zhang & Fu, 2016; Wang et al., 2010). Geodetector enables convenient and accurate exploration of spatial variance and quantification of drivers and is widely used to quantify the drivers and their interactions affecting vegetation cover change (Liu et al., 2021).

(1) Factor detector

The ability of an arbitrary factor to explain the FVC change is measured using the factor detector, and q-statistic was used to measure stratified heterogeneity in FVC change (Wang, Zhang & Fu, 2016). The q takes on a range of values from 0 to 1, the closer the value of q is to 1, indicates greater explanatory power of factor x for FVC change. The corresponding formulas are Chen et al. (2020b) and Liu et al. (2021): (8) q=1−∑h=1LNhσh2/Nσ2

where q is the ability of the factor to explain the FVC change. N is the sample size, Nh is the sample size of factor x in zone h, σ2 is the variance of the regional FVC change, and σh2 is the variance of factor x in zone h.

(2) Interaction detector

This module is used to determine the explanatory power of the FVC change when the two factors interact. First, the q-values were calculated by the two factors as q (x1) and q (x2). Secondly, the q-value reflecting the interaction of the two factors is calculated as q (x1 ∩x2) and compared with q (x1) and q (x2), indicating the type of interaction between the two variables (Peng, Kuang & Tao, 2019). The types of interaction were classified into five groups (Table 2).

(3) Risk detector.

Table 2 Types of interaction between the two factors.

Interaction	Description	
Weaken, nonlinear	q (x1∩ x2) < Min (q (x1), q (x2))	
Weaken, univariate	Min (q (x1), q (x2)) <q (x1∩x2) < Max (q (x1)), q (x2))	
Enhance, bivariate	q (x1∩ x2) > Max (q (x1), q (x2))	
Independent	q (x1∩ x2) = q (x1) + q (x2)	
Enhance, nonlinear	q (x1∩ x2) >q (x1) + q (x2)	

The risk detector was utilized to analyze whether the mean values of the drivers of FVC change differed significantly across sub-regions and to identify the range or type of factor that was most favorable. Risk detection is based on the t-statistic (Liu et al., 2021): (9) t=Y ¯h=1−Y ¯h=2VarY ¯h=1Nh=1+VarY ¯h=2Nh=1

where Y ¯h is the mean of the FVC change in zone h; Nh is the sample size of factor x in zone h, and Var is the variance.

Results

Temporal variation characteristics of FVC change

Figure 2 shows the fluctuating trend of the average FVC from 2000 to 2020. The FVC ranged from 50.26 to 63.45%, with an average of 56.10%. Overall, FVC variation had a rising pattern, with a growth rate of + 0.26/year (P = 0.028), indicating that remarkable progress has been accomplished by implementing ecological protection projects during the last 21 years. The FVC change can be divided into three stages: 2000 to 2005 shows a slight upward trend, 2005 to 2010 shows a downward trend, since 2010 the quality of vegetation has been good but fluctuation is large.

From 2000 to 2020, the FVC of each microtopography in the Muru Basin changed significantly (Fig. 3). The highest FVC is 83.46% for the lower slope (cool), which is significantly higher than other microtopographic types (Fig. 3C). The lowest FVC is 0, distributed in the peak/ridge (cool) (Fig. 3A). The mean FVC for different microtopographic types 2000–2020 is ranked from highest to lowest as follows (Table 3): lower slope (cool) (69.62%), lower slope (62.90%), lower slope (warm) (58.42%), valley (57.39%), upper slope (warm) (54.42%), upper slope (54.15%), valley (narrow) (53.31%), upper slope (cool) (47.13%), cliff (44.42%), mountain/divide (32.77%), peak/ridge (warm) (23.61%), peak/ridge (17.92%), peak/ridge (cool) (2.74%). As shows in Table 4, from 2000 to 2020, the FVC of microtopography types: peak/ridge (warm), cliff, upper slope, lower slope, and lower slope (cool) increased significantly (P < 0.05), while other microtopographic types did not. The microtopography with the fastest growth in FVC is the cliff, lower slope, upper slope, and lower slope (cool), with growth rates of +0.79/year, +0.53/year, +0.52/year, and +0.49/year, respectively (Table 4).

Figure 2 FVC change tendency from 2000 to 2020.

Figure 3 The FVC change on various microtopographic types of Muru Basin.

(A) Peak/ridge, Mountain/divide, and Cliff; (B) Upper slope; (C) Lower slope; (D) Valley.

Spatial variation characteristics of FVC change

Figure 4 shows the spatial distribution of microtopographic types and FVC changes in the study area. The areas with insignificant change, extremely significant increase, significant increase, extremely significant decrease, and significant decrease of FVC accounted for 70.42%, 13.08%, 8.62%, 4.04%, and 3.84% of the whole study region, respectively (Fig. 4B). Most regions with significant changes in FVC showed an extremely significant and significant increasing trend, only a small number of areas have shown extremely significant or significant decreasing trends (Fig. 4B). Figure 5A shows that the study area has the largest number of areas with Hurst exponent less than 0.5 (opposite) (67%), followed by greater than 0.5 (continuous) (19.22%) and equal to 0.5 (random) (13.76%). To further determine the sustainability of vegetation restoration in the study area, the FVC change trend and Hurst exponent were spatially overlaid to analyze the future tendency of FVC change (Fig. 5B). The results showed that the areas from degradation to improvement, continuous increase, and continuous significant increase accounted for 33.35% of the whole area. Random fluctuation areas mostly located at the margin of the basin accounted for 19.17% of the whole area. The areas from improvement to degradation, continuous decrease, and continuous significant decrease accounted for 47.48% of the whole area. This indicates that vegetation degradation is likely to occur in the future.

Table 3 Mean FVC for different microtopographic types 2000–2020.

Microtopographic types	Mean FVC (%)	
Lower slope (cool)	69.62	
Lower slope	62.90	
Lower slope (warm)	58.42	
Valley	57.39	
Upper slope (warm)	54.42	
Upper slope	54.15	
Valley (narrow)	53.31	
Upper slope (cool)	47.13	
Cliff	44.42	
Mountain/divide	32.77	
Peak/ridge (warm)	23.61	
Peak/ridge	17.92	
Peak/ridge (cool)	2.74	

Table 4 Temporal variation characteristics of FVC change.

Microtopographic types	Regression equation	P-value	
Peak/ridge (warm)	y = −0.106x + 236.59	0.327	
Peak/ridge	y = 0.2492x −482.97	0.043	
Peak/ridge (cool)	y = 0.1855x −370.08	0.369	
Mountain/divide	y = 0.1504x −269.52	0.162	
Cliff	y = 0.7926x −1548.7	0.006	
Upper slope (warm)	y = 0.0204x + 13.436	0.327	
Upper slope	y = 0.5232x −997.56	0.001	
Upper slope (cool)	y = 0.4653x −888.14	0.085	
Lower slope (warm)	y = 0.1276x −198.14	0.359	
Lower slope	y = 0.5337x −1009.8	0.002	
Lower slope (cool)	y = 0.4922x −919.62	0.040	
Valley	y = 0.2583x −461.88	0.123	
Valley (narrow)	y = −0.0316x + 116.86	0.894	

Figure 4 Spatial distribution of microtopographic types and vegetation coverage change on Muru Basin.

(A) Microtopographic types; (B) FVC change.

Figure 5 Sustainability of vegetation coverage change on Muru Basin.

(A) Hurst exponent; (B) future trends of FVC change.

It is clear from Fig. 6A that the highest percentage of the extremely significant increase in FVC was the cliff, accounting for 15.57%, followed by the lower slope (warm), lower slope, upper slope, and lower slope (cool), accounting for 15.41%, 14.43%, 14.18%, and 13.73, respectively. It is also clear that the significant increase of FVC was highest in the lower slope, accounting for 12.53%, followed by the upper slope, cliff, and lower slope (cool), accounting for 12.21%, 10.81%, and 10.03% (Fig. 6B). The proportion of extremely significant decrease and significant decrease in FVC on the upper slope (cool) and lower slope (cool) was much less than on the upper slope (warm) and lower slope (warm) (Figs. 6C and 6D). This is likely due to the lower evaporation and better water conditions on the upper slope (cool) and lower slope (cool) than on the upper slope (warm) and lower slope (warm), so the growth of vegetation is stronger.

Figure 6 FVC change trend on different microtopographies.

Future variation trends FVC are disparate in each microtopographic type (Fig. 7). The proportion of the continuous significant increase in FVC is greatest on the lower slope (warm) (Fig. 7A). The proportion of the continuous increase in FVC was greater on the valley (narrow) and valley than on other microtopographic types (Fig. 7C). The area with the largest proportion of FVC from degradation to improvement was the upper slope (warm) (36.78%), accounting for 7.3% of the total study area (Fig. 7E). Compared with other microtopographic types, valley (narrow) FVC showed the highest proportion of continuous significant decrease and continuous decrease (Figs. 7B and 7D). In the lower slope (cool) (69.41%) and lower slope (58.20%), the proportion of FVC coverage from improvement to degradation was larger, accounting for 10.23% of the total study area (Fig. 7F). The future variation trend was a continuous significant increase, continuous increase, or from degradation to improvement indicating vegetation improved. On the contrary, continuous significant decrease, continuous decrease, and from improvement to degradation indicate vegetation degradation. This means that although there is a greater proportion of continuous improvement in FVC in the lower slope (warm), valley (narrow), and upper slope (warm), there is still a risk of vegetation degradation. The proportion of random fluctuation was more extensive in the peak/ridge area (include peak/ridge (cool), peak/ridge, and peak/ridge (warm)) (Fig. 7G), indicating that afforestation in the peak/ridge area was difficult and had a principal impact on the survival of vegetation.

Figure 7 Future FVC change trend on different microtopographies.

Identification of driving forces

(1) Factor detector analysis

We quantified the effect of each factor on the FVC change using Geodetector (Fig. 8). The effect of each factor in FVC change was significant (P < 0.05), except for the slope (P = 0.273). The explanatory power (q-statistic value) of aspect (q = 21.28%, P < 0.001), microtopographic type (q = 8.61%, P < 0.001), soil taxonomy great groups (q = 8.12%, P < 0.001), and soil water content at 200 cm (q = 5.75%, P < 0.001) exceeded 5%, are the main drivers of FVC variation in the study area. These are followed by soil water content at 30 cm (q = 3.77%, P < 0.001), distance to roads (q = 2.90%, P < 0.001), annual average temperature (q = 2.57%, P < 0.001), soil water content at 0 cm (q = 1.71%, P = 0.005), distance to residences (q = 1.57%, P = 0.010), soil pH (q = 1.53%, P = 0.010), soil water content at 10 cm (q = 1.41%, P = 0.019), and slope (q = 0.66%, P = 0.273). This indicates that aspect, microtopographic type, and soil taxonomy great groups are important factors influencing FVC change in the study area. Although human activities such as roads and residences are important factors, in our study area they are much less important than natural factors such as aspect. This probably is due to the implementation of policies such as the NFPP and RCFGP have greatly reduced the disturbance of vegetation by human activities. In addition, the vegetation in the study area is mainly deep-rooted vegetation such as Abies squamata, Picea likiangensis. Therefore, soil water content at 0–30 cm did not show a significant explanatory power to the variation of FVC.

(2) Interaction detector analysis

Figure 8 The explanatory power of factors for FVC change.

** indicates P < 0.01 and * indicates P < 0.05.

Figure 9 shows that the interactions between the two factors all have stronger explanatory power for the FVC changes than the individual factors and that the majority of the interaction effects are non-linearly enhanced. Specifically, the interactions between aspect and microtopographic type, soil taxonomy great groups, soil water content at 200 cm; and the microtopographic type and soil water content at 200 cm; and soil water content at 30 cm and soil water content at 200 cm demonstrated bivariate enhancement, whereas the interaction between other factors showed a nonlinear enhancement. According to the results, the q-statistic value of the interaction between any factor and aspect was larger than that of the other two factors, indicating that aspect played an important role in FVC change. This was consistent with the conclusion in factor detector analysis, the q-statistic value of aspect was 21.28%. In addition, amongst these interactions, aspect and annual average temperature (q = 29.20%) were the strongest. Although the explanatory power of annual average temperature for FVC change is less than 3% (q = 2.57%), its interaction with aspect has larger explanatory power than the sum of each and is much larger than the interaction between the other factors. It can be seen that annual average temperature also has a significant effect on the change of FVC, mainly in the interaction with aspect.

(3) Risk detector analysis

The risk detector can detect the most suitable level of vegetation restoration for each factor. A higher value of FVC slope for a given level represents a more favorable level of this factor for vegetation restoration, with different factors presenting significant differences in the value of FVC change (Fig. 10). The response of the FVC slope to the aspect of different levels first decreases and then increases (Fig. 10A). The FVC slope was the highest when the aspect was 292.5∼337.5° (northwest), 337.5∼22.5° (north), and 22.5∼67.5° (northeast) (Fig. 10A), indicating that the northern slope (292.5∼67.5°) was the most suitable for vegetation restoration. Concerning the microtopographic types, the lower slope (cool) was conducive to vegetation restoration, with the largest FVC slope (Fig. 10B). The soil taxonomy great group most conducive to vegetation restoration was Cryumbrepts (Fig. 10E). Cryumbrepts are well- or moderately drained soil common in subalpine areas that support forest, grassy, and tundra vegetation (Rieger, 1983). The value of the FVC slope increases with the increase of soil water content at 200 cm depth (Fig. 10I). When the water content at 200 cm was 15∼20%, it was the most suitable for vegetation restoration (Fig. 10I). The FVC slope was highest at an annual average temperature of 1.52∼3.14 °C (Fig. 10J), which was the most favorable level for vegetation restoration. In terms of the impact of human activities on vegetation recovery, the FVC slope shows an increase followed by a decrease with increasing distance to roads and residential areas, with the greatest FVC slope in the range of 1500∼2000 m from roads (Fig. 10K), and the greatest FVC slope in the range of 1000∼2000 m from residences (Fig. 10M).

Figure 9 Interaction effects of different factors on FVC change.

Numbers indicate explanatory power, “↑” indicate Enhance and nonlinear, “↑ ↑” indicate Enhance and bivariate.

Figure 10 Results of FVC change for each class of different factors.

For easy observation, the value of the FVC slope was increased by 10,000 times.

Discussion

Vegetation change features and the impact of microtopographic type

Overall, there has been an increasing trend in FVC on the Muru Basin from 2000 to 2020, which is consistent with a positive trend in vegetation cover on the TP (Song, Jin & Wang, 2018; Zhang et al., 2018). However, it must not be overlooked that some of the microtopographic types are degraded (Figs. 5 and 7), where vegetation restoration is not sustainable. Due to extreme climatic conditions and ecosystem fragility in the subalpine areas of the ETP, the ecology is highly sensitive to logging activities, and post-harvest ecological restoration faces enormous challenges (Xiong et al., 2021). Even some achievements have been made in forest restoration by government support, but the survival rate of afforestation is still low in some areas (He, Shi & Fu, 2021), and vegetation is degraded in some places (Wang et al., 2019). The uncertainty in vegetation restoration is a major challenge for sustainable forest management and conservation in subalpine areas.

We found that the lower slope was the microtopographic type with the best vegetation cover (Fig. 3). The ridge/peak areas had the greatest proportion of FVC decrease and random fluctuation (Figs. 6C, 6D and 7G). Topographic factors (elevation, slope, aspect, and microtopographic type) play an essential role in the dynamics of vegetation cover (Jia et al., 2020; Li, Shi & Wu, 2020). They influence the climate and soil moisture conditions during the year by controlling precipitation, solar radiation, and temperature (Metzen et al., 2019). The lower slopes have a non-native water input from upper slope areas in addition to groundwater (Thompson et al., 2011), which is more beneficial for plant growth. The lower slope has better hydrological conditions compared to the other microtopographic types (Thompson et al., 2011), and in study areas where Abies squamata and Picea likiangensis are common forest species, moist soil are more conducive to their growth. The ridge/peak areas soils are relatively shallow, soil moisture and nutrients are lacking (Gao et al., 2017a), so afforestation is difficult. In general, relatively humid lower slopes are the most suitable for vegetation restoration, and peak/ridge should be avoided as much as possible.

Key factors influencing vegetation restoration

The morphology and function of watersheds result from long-term co-evolution between water, soil, landforms, and ecosystems (Thompson et al., 2011), with solid feedback between vegetation, topography, climate, and soil (Kirkpatrick et al., 2014). Our study used Geodetector to determine the key drivers of FVC change in subalpine regions. In contrast to other statistical approaches, it can measure Spatial Stratified Heterogeneity (SSH) and analyze the effects of factors and their interactions on FVC (Liu et al., 2021).

Our research shows that aspect is the most critical factor influencing FVC change, followed by the microtopographic type and soil taxonomy great groups (Fig. 8). Previous studies have also shown that topography and soils are important drivers of vegetation restoration (He, Shi & Fu, 2021; Zhong et al., 2022). The northern slope (292.5∼67.5°) was the most suitable for vegetation restoration (Fig. 10A); they are more humid than the southern slopes (Qingkong et al., 2020; Sun et al., 2019). Predecessors’ research supports our conclusion that northern slopes have adequate soil moisture and higher vegetation cover than southern slopes (Louhaichi et al., 2021; Metzen et al., 2019). The lower slopes are recharged by both groundwater and non-native water from the upper slopes (Thompson et al., 2011), and are more suitable for vegetation growth. In addition, soil texture also influences FVC change, with the most suitable soil taxonomy great groups for vegetation restoration being Cryumbrepts. Soil is an important determinant of vegetation restoration (Mills et al., 2021), affects the growth and composition of vegetation (Fernández-García et al., 2021). Cryumbrepts are consistently well- or moderately drained soil (Rieger, 1983), well-drained soils are more conducive to vegetation root expansion, respiration, and growth (Crowell & Lane, 2001). In vegetation restoration, the drainage of the planting site is essential for the health and growth of seedlings.

Roads and residences are the concrete embodiment of human activities in the study area, they are mainly located in lower elevations and valleys areas, and the residences are mostly located along the roads. The field survey found that areas close to roads and residences in the study area are mostly cultivated and heavily influenced by human activities, which is not conducive to vegetation restoration. Further away from the roads and residences are located at the edges of the basin and are meadows where the vegetation is more affected by grazing and self-growth restrictions. Accordingly, the results of the study show that areas moderately close to roads and residences are the most favorable for vegetation restoration. However, the contribution of distance from roads and distance from residents to FVC change is only 2.90% and 1.57%, respectively, which is much less than 21.28% for aspect and 8.61% for microtopographic types. Since 1998, a series of ecological protection projects have been implemented, such as NFPP and RCFGP, which have greatly limited human disturbance and damage to the local ecology. Overall, the impact of human activities on vegetation restoration in the study area is much less than the natural factors.

Besides, the diversity of geographical processes shows that the interplay of various factors influences changes in vegetation cover (Huo & Sun, 2021). Interaction between the two factors had stronger explanatory power for the FVC changes, which is similar to the idea that the interaction between various factors has a markedly stronger effect on the vegetation than the individual factors (Liu et al., 2021). We explored that annual average temperature also has a significant effect on the change of FVC, mainly in the interaction with aspect. In particular, the interaction between temperature and aspect has the most excellent effect on FVC change, probably because aspect variation leads to differences in temperature (Pepin et al., 2017), which affects the vegetations’ spatial distribution and growth.

Limitations and uncertainties

Vegetation restoration is influenced by a combination of topography, climate, soils, and human activity (Chen et al., 2020b; Meng et al., 2019; Peng, Kuang & Tao, 2019). Our study used Landsat imagery with a resolution of 30 m to capture the dynamics of vegetation restoration. Unlike MODIS, Landsat with higher resolution captures more spatial detail (Seto et al., 2004). However, the 16-day revisit time and frequent cloud cover make it difficult to obtain sufficient high-quality data, which may reduce its performance in detecting rapid ecosystem change (Liao et al., 2016). A spatial–temporal fusion approach to mix MODIS and Landsat into new composite data seems to be a good solution. In this study, the main drivers of vegetation restoration in subalpine areas were identified and analyzed using Geodetector, which incorporates a range of spatial statistical methodologies to explore the explanatory variables affecting the dependent variable through a spatial variance analysis (Wang, Zhang & Fu, 2016; Wang et al., 2010). Nevertheless, some of the factors such as precipitation and elevation, which have a strong influence on vegetation restoration (He, Shi & Fu, 2021; Zhang et al., 2021), are excluded due to severe multicollinearity. This is most likely caused by the small scale of our study area and the need to consider a more detailed study on a larger scale such as the ETP.

In addition, previous research has shown that the impact of human activity on vegetation change is crucial (Cheng et al., 2021; Meng et al., 2019). Whereas, our study found that distance from roads and distance from residents contributed only 2.90% and 1.57% respectively to the change in vegetation cover. This indicates that the impact of human activity on vegetation in our study area is much lower than other natural factors. On one hand, the implementation of ecological projects limited the impact of human activities. On the other hand, it would be because the roads and residents could not cover all human activities in the study area. Some studies use land-use data to measure the intensity of human activities and have achieved good results (Chen et al., 2020b). We can combine roads, residents, and land-use data to quantify the intensity of human activities for further research.

Conclusions

This study used trend analysis and the Hurst exponent to explore the spatial and temporal variation and future sustainability of vegetation cover in the Muru Basin from 2000 to 2020, and quantified the influence of various drivers and their interactions on FVC changes based on Geodetector. Regional vegetation cover has shown an increasing trend since 2000. Aspect, microtopographic type, and soil taxonomy great groups are the main factors influencing vegetation cover change in subalpine regions. The lower slope has the highest mean FVC, peak and ridge have a relatively low mean FVC. The northern slope (292.5∼67.5°) is the most suitable aspect for vegetation restoration. Accordingly, we suggest selecting lower slope, northern slope, and better-drained soils (Cryumbrepts) for vegetation restoration, and areas such as peak and ridge should be avoided as much as possible for afforestation. The implementation of a series of ecological projects in the study area has reduced the impact of human activities, and regional vegetation restoration is mainly limited by natural factors, with a low impact of human factors. The study results enrich the understanding of vegetation cover changes within a typical basin in the subalpine region of the ETP and reveal important factors affecting vegetation restoration. The results of the study provide theoretical references and suggestions for vegetation restoration and sustainable development in typical logging areas in the subalpine region. However, we need to take the potential drivers of vegetation restoration more comprehensively in future studies.

Supplemental Information

Supplemental Information 1 The code to calculate the FVC in GEE

Click here for additional data file.

Supplemental Information 2 FVC slope

Click here for additional data file.

Supplemental Information 3 FVC Hurst

Click here for additional data file.

We thank the openbiox community and Hiplot team (https://hiplot.com.cn/basic?lang=en) for providing technical assistance and valuable tools for data analysis and visualization.

Additional Information and Declarations

Competing Interests

Author Contributions

Data Availability

Qin Zhou is employed by Chengdu OCI Medical Devices Co., Ltd. The authors declare there are no competing interests.

Yu Feng conceived and designed the experiments, performed the experiments, analyzed the data, prepared figures and/or tables, and approved the final draft.

Juan Wang and Peihao Peng conceived and designed the experiments, authored or reviewed drafts of the paper, and approved the final draft.

Qin Zhou performed the experiments, prepared figures and/or tables, and approved the final draft.

Maoyang Bai performed the experiments, analyzed the data, prepared figures and/or tables, and approved the final draft.

Dan Zhao analyzed the data, authored or reviewed drafts of the paper, and approved the final draft.

Zengyan Guan and Xian’an Liu analyzed the data, prepared figures and/or tables, and approved the final draft.

The following information was supplied regarding data availability:

The code and raw data are available in the Supplementary Files.

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
