# Peer review of "Quantitative analysis of vegetation restoration and potential driving factors in a typical subalpine region of the Eastern Tibet Plateau"

_PeerJ, doi:10.7717/peerj.13358_

## Round 0.1 · original submission · Major Revisions

Reviewers' comments on your work have now been received. The manuscript has been assessed by two reviewers. Reviews indicated that the Methods, Conclusion, Discussion etc. should be improved. In addition, the influence of human activities is not considered in the potential drivers of this paper. I agree with this evaluation and I would, therefore, request for the manuscript to be revised accordingly.

Reviewer 1 ·

Basic reporting

no comment

Experimental design

The theme of this paper is "Quantitative analysis of vegetation restoration and potential drivers in typical subalpine areas of the eastern Tibetan Plateau", but most of the research in this paper focuses on the study of vegetation cover and environmental factors. However, vegetation restoration is a system project, which is influenced by many factors, among which human activities have a greater influence on vegetation, and the influence of human activities is not considered in the potential drivers of this paper, which is also inappropriate.

Validity of the findings

no comment

Additional comments

no comment

Reviewer 2 ·

Basic reporting

1. You should give more details for the study area introduction, and let the reader know more about the area.
2. You should add the lat and lon in the figure 1, and let the reader know the specific location.
3. Have you estimate the VIF for those driving factors? I think there are VIF to some extent.
4. In the section of conclusion, you give a lot of results description, but not conclusions.
5. In the section of discussion, you had better give some details for the limitation and uncertainty of this study.
6. There are many small errors, you should check the whole manuscript. For example, km2 in line 113.

Experimental design

No comment

Validity of the findings

No comment

Additional comments

No comments

---

## Round 0.2 · Major Revisions

Reviewers' comments on your work have now been received. The manuscript has been assessed by three reviewers. Reviewers indicated that the introduction, data sources, and results should be improved. Moreover, the available raw data should be provided. Moreover, English editing and academic writing service are needed. I agree with this evaluation and I would, therefore, request for the manuscript to be revised accordingly.

Reviewer 5 has requested that you cite specific references. You may add them if you believe they are especially relevant. However, I do not expect you to include these citations, and if you do not include them, this will not influence my decision.

Reviewer 3 ·

Basic reporting

The authors present a very important problem of vegetation recovery in subalpine areas. Generally, the manuscript seems to be within the scope of the PeerJ. In my opinion, the article is very interesting and does not require many corrections, but an improvement of the work is needed to give this paper its important value.
Point 1: Introduction (Line 91-99) could be described more precisely for the possible drivers of vegetation change across the Eastern Tibetan plateau, at least the Tibetan plateau, which may be more meaningful for you work.

Experimental design

Point 2: Data sources (Line 131-137): How to deal with the cloud influence about Landsat satellite images obtained from Google Earth Engine. How many images do you get for each pixel each year. It will be better to add a frequency map of images information obtained.

Point 3: Results (Line 294): Why do you make a separate section about FVC change on different micro-topography. I mean it could be include in the spatial variation characteristics of FVC change section. I propose to add a more detailed description about this objective or combine them.

Validity of the findings

Point 4: Which do you think is more important for vegetation change in your study area, natural or human factors? Some discussions about how to distinguish them could be added.

Additional comments

Point 5: There need some grammar check, such as line 415 comma.

Reviewer 4 ·

Basic reporting

Specific details:
1. The title format of each chapter in the manuscript needs to be modified based on the submission guidelines.
2. Inconsistent use of some terms. The descriptions of “slope(warm)” and “slope(cool)” are irregular. And in Line 303-304, the description becomes “sunny slope”. Please standardize academic terms.
3.Please simplify Figure 3. The information in Figure 3 is too much.

Experimental design

no comment

Validity of the findings

4.In Line 257-260, how do the study come to this result? And what year of the data is used?
5. In Line 317-326, in addition to the concise and precise description of results, there should also be a brief analysis of this phenomenon.

Additional comments

This study analyzed the temporal and spatial characteristics of fractional vegetation cover (FVC) in a typical subalpine region based on linear regression and Hurst exponent. Furthermore, the driver factors of FVC were identified by Geographical Detector. The manuscript is well written, and the results can provide a scientific basis for future research. Some ideas deserve attention, but further revisions are surely needed.

Reviewer 5 ·

Basic reporting

I have completed the reviewed the manuscript peerj-68087 Title: Quantitative analysis of vegetation restoration and potential driving factors in a typical subalpine region of the Eastern Tibet Plateau.
First of all, I affirm the positive response and improvement of the manuscript to the opinions of the two reviewers
I though this work can be given us a case-reference in understanding of vegetation restoration changes within typical watersheds in the subalpine region of the ETP and reveal important factors affecting vegetation restoration. The significance, relevance and timeliness of the topic was well done. However, this manuscript has some sections and expressions need to improve. For example, (a) quality and content of the research/review . Comments: Some details need to be improved.
(b) quality, brevity and clarity of presentation . Comments: Some parts need to improve.
(c) Innovation points, key links of research methods and uncertainty analysis of research results need further improvement.
Hence, on the basis of all my following comments, the publication of the manuscript in its present form is not recommended, and minor revision is requested.

Experimental design

First, It is recommended to provide a logically technical framework of your study how to be done, and a list of relevant information about the main data, and the analysis codes on GEE platform, which can be in the form of Appendix (i.e.,Supporting information).
Second, fractional vegetation cover (FVC) is the key word of this study, but important ecological background information such as vegetation composition and spatial pattern in the study area before and after vegetation restoration is not fully described in the manuscript, therefore, it is suggested to supplement relevant information from the perspective of phytocoenology.
Third, deciduous vegetation is involved in the research object, whether the influence of seasonal differences on FVC is considered.
Lastly, there are some the key research methods, especially the original ones of yourself, need to be explained clearly and their literature. eg, About human activities, Why did not you only choose distance from roads and distance from residents as explanatory variables for FVC changes?Why did not you consider soil nutrients, soil moisture, land use and other factors in higher spatial resolution?
In addition, there are many relative researches closely related to your research, it is recommended to make a search, read and reference the following literature.
【1】Fengkui Ma, Qun’ou Jiang, Lidan Xu, Kexin Lv, Guoliang Chang. 2021. Processes, potential, and duration of vegetation restoration under different modes in the eastern margin ecotone of Qinghai-Tibet Plateau. Ecological indicators, 132:108267, doi.org/10.1016/j.ecolind.2021.108267
【2】Xinping, Zhang , et al . 2017. Effects of land use/cover changes and urban forest configuration on urban heat islands in a loess hilly region: case study based on Yan’an city, China. International Journal of Environmental Research and Public Health, 14(8), 840. doi.org/10.3390/ijerph14080840.
Supporting information Section
【3】Zhang Xinping, Zhang Fang Fang, Wang Dexiang, et al. 2018. Effects of vegetation, terrain and soil layer depth on eight soil chemical properties and soil fertility based on hybrid methods at urban forest scale in a typical loess hilly region of China. PLoS ONE 13(10): e0205661. doi.org/10.1371/journal.pone.0205661

Validity of the findings

(1) Please explain why only vegetation coverage was selected as a measure of vegetation restoration effectiveness.
(2)The minimum non-heterogeneous spatial unit and its scale need be explained in detail.
(3)In the section of Limitations and uncertainties, the analysis and explanation in the current version of manuscript are not scientific enough, it is suggested to make a quantitative analysis of the uncertainty of the influence factors affecting FVC.
(4)In the conclusion part, the summary of the research results is weak, and Lack of recommendations for further research.

Additional comments

(1)English expressions:
I noticed that your English usage is slightly off and could benefit from editorial help of a fluent English speaker or editorial service.
(2)Novelty: Unclealy, please further clarify the innovation of your research.
(3)Finally, I suggest that the authors carefully examine the title of the article.

---

## Round 0.3 · accepted · Accept

Reviewers' comments on your work have now been received. The manuscript has been assessed by two reviewers. Reviewers indicated that the authors have addressed all the issues in the previous round of review.

Reviewer 3 ·

Basic reporting

no comment

Experimental design

no comment

Validity of the findings

no comment

Additional comments

no comment

Reviewer 4 ·

Basic reporting

no comment

Experimental design

no comment

Validity of the findings

no comment

Additional comments

no comment